# Why Is the Spanish Hotel Trade Lagging So Far Behind in Gender Equality? A Sustainability Question

**María Jesús Carrasco-Santos** [1,2,*] **, Carmen Cristófol Rodríguez** [3] **and Eva Royo Rodríguez** [4]

1 Department of Economics and Business Administration, Faculty of Tourism, University of Málaga, 29010 Málaga, Spain
2 Institute of Tourist Investigation, Intelligence and Innovation, University of Málaga, 29010 Málaga, Spain
3 Faculty of Business and Communication, International University of La Rioja, 26006 Logroño, Spain; carmen.cristofol@unir.net
4 Faculty of Tourism, University of Málaga, 29010 Málaga, Spain; evaaa.rr317@gmail.com
* Correspondence: mjcarrasco@uma.es

**Abstract:** There are far more women than men who hold higher qualifications in tourism, but nevertheless, inequalities still prevail on executive committees in the tourism industry. Society nowadays is aware that gender inequality in terms of women's rights and opportunities has always existed. Such problems are reflected in religious doctrines, cultural habits, and outdated ways of thinking in which women are viewed as being first and foremost careers. This attitude is also reflected in the workplace. Methodology: Hakim's (1992) methodology has been used to classify occupations, and female-dominated, male-dominated and integrated occupations can be used, studying data published by the most important hotel chains in Spain. Objectives: since women joined the labor market, females have been employed in significant numbers in the tourism sector. The objective of this work is to study in depth what number of women are in the boardrooms of large hotel chains in Spain, collecting data from the top five hotel chains, as this important phenomenon must be researched. The most salient results are: a high number of women work in the hotels subsector and gender equality has not yet been achieved at higher echelons, since senior management positions are dominated by males. Hence, the aim of this paper is, having carried out a thorough and extensive evaluation, to empirically determine the state of play for females in this industry today and to put forward further improvements which women, the government, hotels and society should jointly strive to achieve. In the conclusion, the initial hypothesis is confirmed.

**Keywords:** woman; qualifications; management; tourism; discrimination; workplace; gender; leadership

## 1. Introduction

"Women first" is a cry often heard, a call to protect the supposedly "weaker sex" against the stronger one and an attitude which is still deeply embedded in the mindset of many people today. Questions might be posed such as: what advantages do women really have in today's society? What role do they play in business? Does discrimination and, consequently, talent loss exist? The aim of this research is to answer these questions by using different sources of research in order to demonstrate that top management positions in the tourism-hotel trade are fundamentally occupied by men, and to, in turn, gain an understanding of why this situation persists so as to come to some conclusions supported by statistical evidence, Hakim's (1992) [1] methodology has been used. Moreover, there has been little research in the tourist industry about this on a nationwide scale [2–4]. In fact, gender equality in employment remains a controversial issue in society and many matters are yet to be resolved.

Therefore, the main reason underpinning this research is to assess the degree of gender inequality in terms of top management positions in the boardroom, specifically looking at the tourism industry, and to focus on the largest subsector within this industry: hotels. Gender is understood to be all beliefs, personality traits, attitudes, feelings, values and socially constructed behavior which set men apart from women [5].

We have broken our main objective down into the following specific ones:

- To measure to what extent women participate in the tourism and hotel industry in order to assess whether discrimination exists or not, by examining a sample of members of boardrooms of hotels in the main hotel chains in Spain.
- To draw some conclusions about the potential implications of these findings and to make some recommendations.

This phenomenon is known by many authors as the "glass ceiling" or "GC" which refers to the barriers women face to obtaining executive positions, or as Durbin [6] defines it, this is the result of a patriarchal hierarchy in which men whether individually, as a group or through the force of argument, exercise power over women. This concept is covered in hotels by analyzing different determining factors in this situation.

As established by the General Office for Studies, Analysis and Action Plans from the Ministry of Industry, Energy and Tourism [7] and The General Office for Coordination and University Monitoring from the Ministry of Education, Culture and Sport [8], in the academic year 2015–2016, 8342 men and 17,285 women enrolled in Travel, Tourism and Leisure studies. Additionally, in recent years at some Tourism Schools, the average marks gained by women have been higher than those gained by men, so generally speaking they are better qualified [9]. In light of this finding, the following question may be posed: why are there so few women in the hotel trade who are on a board of directors and who hold executive positions? How does this affect hotels? Do all women face the same barriers to accessing high responsibility positions in these companies? Studies have been carried out which have shown women to be less valued than men, even when performing the same work [10,11].

In short, the employment position of women in the hotel trade is in need of reform and this would benefit society as a whole and the reason why change is so crucial will be explained later on in this paper. Segovia and Figueroa [12] take the view that the phenomenon of women being less represented in executive positions than men is a matter of social justice; moreover, it is tantamount to talent loss, although other research shows talent management rather than gender equality may be required to remedy this situation.

## 1.1. Special Features of the Tourism and Hotel Industry

As already demonstrated by all the contributions made in previous research, women are less prevalent than men in high responsibility positions, which therefore shows that they also face barriers to promotion to executive positions. From the point of view of Molpeceres, Ongil, Henar and Rodríguez [13], initiatives and strategies for promoting women to high responsibility positions can be broken down into three main areas: the attitude women themselves have, reform of business organizations and promotion of a work-life balance. In fact, these authors view the tourism sector to be in pressing need of reform because their research has led them to the conclusion that: "tourism provides the starkest imbalance between the overall level at which women are represented in their occupation and their representation at professional and executive echelons".

So, what are the special features of the tourism and hotel industry? It could be described as an economic activity which requires total commitment, as its main reason to exist is to provide customer service. Moreover, as the main subsector in tourism is accommodation and catering, working schedules are 24-7-365 days of the year, which sets it apart from normal working days. In other words, it transcends so called "office hours". The International Labor Organization (ILO) [14] alludes to the fact that some negative points which working in hotels and restaurants entails are long working hours, standing for

practically the whole day, many hours walking, highly repetitive movements, painful positions, carrying heavy weights, pressure and stress. As for employee conditions, do they fare well in relation to their counterparts in other industries? [2]. Also, added that employment in tourism is associated with lower quality work than other industries, lower salaries and temporary and part-time work.

So, how do these special features of the tourism industry affect female directors? Segovia and Figueroa [12] define the following factors influencing the "Glass Ceiling":

- Internal: These are ones which women can choose, such as level or type of training, family-related factors and the role the woman performs. It covers human capital (age, training and professional experience) and family issues.
- External: These are the ones which have a direct influence on women´s careers or over which women have less decisions-taking capacity because they stem from her cultural setting, and include sociocultural factors, corporate culture and corporate policies.

What age range and professional experience do female directors have? What are the barriers or limits which really make it difficult for them to access these positions? In the report Mujer y turismo: la igualdad no existe (Women and tourism: equality does not exist) it was stated that: "You might think that the employment situation of women is now regularized in comparison with past years, but evidence shows this is not the case: there are still far more male hotel directors than female ones and women are very scarce on boards of directors of tourist companies" [9].

What age range predominates for female hotel directors? According to the [15] EPA (Spanish Active Population Census) from 2015 up until the last quarter of this year, the most widespread ages for female occupation ranged between 25 and 39 years old, and in recent months women aged from 30 to 39 have been more prevalent, which seems to match the profile of female hotel directors. However, the EPA [15] does not specify corporate positions, nor does it expressly refer to the age range of female hoteliers in management positions. The closest approximation we have to this is in the Madrid area, in the study, Techo de Cristal en el Sector Turístico/The Glass Ceiling in the Tourism Industry, in which the average age of female hotel directors in the region is 39 and the mode (the most frequent) is 36. A total of 73% of those in the survey were between 24 and 43 and 25% were aged between 24 and 33. These results indicate that women have only recently been given executive positions [16].

Is training a determining factor when obtaining an executive position? How significant is this in hotels? What about the tourism industry in general? Many authors agree that training levels of women have increased in recent years, and they are higher than men's [9,17]. They do not consider this to be an obstacle to their career development, but quite the opposite [18]. In reality, the main obstacles to reaching gender equality are gender roles, the influence of self-imposed barriers and the difficulty in achieving a work-life balance [19].

Ref. [2], in their study Mujeres directivas en la industria turística: factores determinantes/Female directors in the tourism industry: determining factors in their career, using a sample of 30 female hotel directors throughout Spain, confirmed that:

"Practically all women who have accessed executive positions are very highly qualified and hold university degrees, master´s or MBA (Masters in Business Administration) and all of them keep on training throughout their careers with life-long learning. Language learning is a particularly important part of their training, obviously due to the nature of the tourism industry".

This study also responds to the previous question as training is not an important factor in promotion in the hotel trade; it is patently taken as a given, and must be supplemented with other abilities such as inter-personal skills and flexibility.

So, does the work-life balance exercise more of an influence? Many women face dilemmas and contradictions which force them to recreate their traditional homemaker roles and renounce work either partly or completely, in order to commit themselves to their families and homes [18]. In fact,

Segovia and Figueroa [12] state that the family duties of women have always been considered to be maternity, raising children and general housework, and this often means men are viewed as more valuable employees than women, which explains why they are promoted faster. Furthermore, they stress issues such as the double work load, negotiating housework and the domestic tension which arises when women work outside the home.

The family-work conflict also affects salaries. In Spain, just like in the EU, the rate of female employment falls according to how many children she has, and goes from 77% for childless women down to 52% when she has three or more children [20]. However, the rate of male employment is not affected by paternity. On the contrary, males with three or more children tend to occupy higher positions than childless ones. There tends to be a higher rate of employment among female university graduates (even if they have children under six years old) whose employment rate is around 80% [20]. Furthermore, executive positions are more demanding in terms of hours worked or the amount of travel required, which means many women cannot fulfil their duties to their families. Women are assumed to be more family-orientated and less competent than men, although thanks to tourism and the fact that more women now work outside their homes, traditions in some cultures have changed, which has improved opportunities for women in socio-economic terms [21].

Out of all the internal factors, the most insurmountable one for promoting women to management positions seems to be the work-life balance. Women tend to be more dedicated to the home and their children, and family obligations are some one of the biggest stumbling blocks to their promotion and commitment to the company they work for [12]. Traditional gender roles which associate women with cares still hold sway in corporate environments [22] Moreover, in a study on the roles women have had in Spanish fiction, women still play traditional roles of customer service providers and cares, and should they play executives, they are portrayed as villains [23]. In Spain, women mainly occupy low skill positions in the hotel trade [18].

Durbin [6] believes this is because women have to prove themselves before they can climb the corporate ladder, as companies can make more informed decisions if they know beforehand how women perform and their worth.

## 1.2. Legal Framework for Gender Diversity

From this information it might be said that legislation on the legal framework for gender equality in different regions has foreseen enough laws to address this issue and to provide protection against discrimination. The LOIMH [24] (Law on Equality between Men and Women), has been lauded as a landmark in this respect since 2007. This law has helped improve the situation of women in the workplace, albeit not completely. This can be deduced from the few inspections made to check that the Equality Plan is actually being implemented, which is rare in hotels.

It was seen that companies in this field base their equality policy on CSR and "Sustainability". They were eager to embrace gender equality as a way of improving their public image; they set up Equality Plans but, did not tend to follow them up with reports on what the real results were from such plans. However, it is not obligatory to do this, and there are no penalties for companies which do not comply with such plans. As a result, progress has been slow over the 12 years since the LOIMH (Law on Equality between Men and Women) [24] came into effect. Lastly, balancing work and one's private and family life is what most influences the law when establishing more effective equality at work.

Sustainability in tourism is linked to the preservation of ecosystems, the promotion of human well-being for employees, tourists and residents of the destination [25]. There are a significant number of authors who have analyzed the role of women in social, economic and environmental sustainability, as can be seen in Table 1.

Environmental sustainability has a significant positive association with the companies run by women [26].

**Table 1.** Sustainability and Women.

| Authors | Women Directors/Sustainability |
|---|---|
| Adams and Funk [27]. | Less oriented towards power, more towards understanding the protection of people and nature |
| Adams and Ferreira [28]. | Promote social welfare + protect environment |
| Ahern and Dittmar [29]. | Promoting social, economic and environmental sustainability |
| Bear, Rahman and Post [30]. | Promoting positive behavior on philanthropic, charitable and ethical issues |
| Davidson and Freudenburg [31]. | Women directors are more responsive towards the environment |
| Fernando, Sharfman and Uysal [32]. | Promoting social, economic and environmental sustainability |
| Galbreath [33]. | Improving social, economic and environmental sustainability |
| Groysberg and Bell [34]. | Social welfare |
| Jain and Jamali [35]. | Women directors are more responsive towards the environment |
| Liao, Luo and Tang [36]. | Social, environmental and economic sustainability |
| Liu [37]. | Women less violate the environment/reduce risks/care for people and society |
| Nadeem, Zaman and Saleem [38]. | Welfare of the people, society and the environment |
| Oba and Fodio [39]. | Social, environmental and economic sustainability |
| Post, Rahman and McQuillen [40]. | Positively influence renewable energy alliances |
| Rodriguez–Fernandez [41]. | Promoting social, economic and environmental sustainability |
| Zahid, Rahman, Ali, Khan, Alharthi, Qureshi and Jan [26]. | Women directors have an imperative role in improving Corporate Sustainability Disclosures as evident by their significant positive association with workplace and social sustainability, also the environmental sustainability. |

Source: Prepared by the author according to Zahid, Rahman, Ali, Khan, Alharthi, Qureshi and Jan [26].

Rodriguez–Fernandez [41] explain how there are prejudices towards candidates for leadership positions due to the perception of an inconsistency between the gender role and the role of leader. The sum of prejudices and stereotypes gives rise to discriminatory behavior that causes inequalities in the workplace.

On the contrary, when the woman decides to break the glass ceiling and fight to move up the professional ladder, there will be many efforts that need to be made, the level of demand being higher and sacrificing the role and expectations that her children or spouse may have towards her, which could affect their health [42].

Women suffer the tensions and conflicts of trying to reconcile the two types of bonding (affective dominance and rational predominance) within the workplace, the education that predominates for women is the affective one, which would be what prepares them for the domestic sphere in as opposed to masculine, rational preparation that is more commonly focused on the labor field. Thus, the woman faces a lack of family social support (her family members demand that she reproduce the traditional role of mother), labor (the professional is measured by her dedication and experience), economic (the market imposes their needs linked to capital and women still seem to be a less profitable investment in the medium-long term than men) and even psychological, since the same woman wonders about what she wants and what she needs, causing multiple internal conflicts. If the educational level determines the position they occupy in companies, women should occupy most of the managerial positions, since the majority are women with higher education levels. As stated, however, it must be assumed that there are other determinants that prevent this premise from being fulfilled. Currently, the managerial positions, occupied mainly by men, demand a great availability of hours and employers are reluctant when it comes to employing women in these positions due to the fact that they assume the

bulk of family responsibilities, mainly childcare, as well as the negative effects produced by being in a double presence in the domestic and work sphere that finally make some of them abandon the job [21].

Also, women suffer from a vertical segregation, which refers to the discrimination suffered by women in their professional careers, in relation to job promotion. This type of gender discrimination in employment has been illustrated through the glass ceiling metaphor, which highlights the invisible barrier that prevents women from reaching positions of maximum responsibility and business leadership, the glass ceiling is a problem that can be: self-imposed barriers, institutional or organizational [19]; also, one of the main barriers that prevent women from accessing management positions in this type of institution is related to the existence of a dominant «patriarchal worldview» in Spanish society and culture [43].

Figure 1 describes the types of barriers identified for the existence of individual, cultural and institutional barriers.

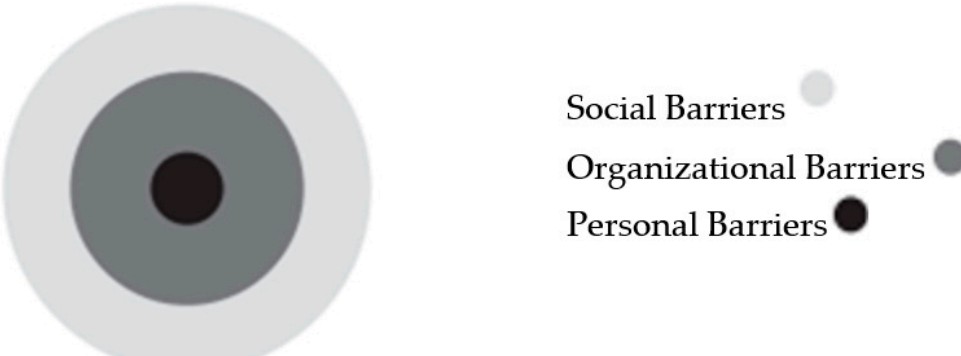

**Figure 1.** Barriers forming the glass celling; Source: Gaete-Quezada [43].

Personal barriers: Aspects related to the personal or family life of women, which affect or condition their own options to access managerial positions.

Organizational Barriers: Situations present in the organization in which women develop their work careers, which hinder their access to managerial positions.

Social Barriers: Distinctive aspects of the culture of the society or the territory where the labor organization works.

One personal barriers is motherhood and another is the problem of reconciling work with the family, especially when more time is required in managerial positions, as can be seen based on Grounded Theory [44], which identifies that some women need to condition a balance between work and family responsibilities. Its main conclusion is that women in managerial positions seek the necessary support in their private sphere in order to develop their jobs in this way, manifesting a barrier to the careers of the women.

Figure 2 describes relevant aspects involved in reconciling work and family.

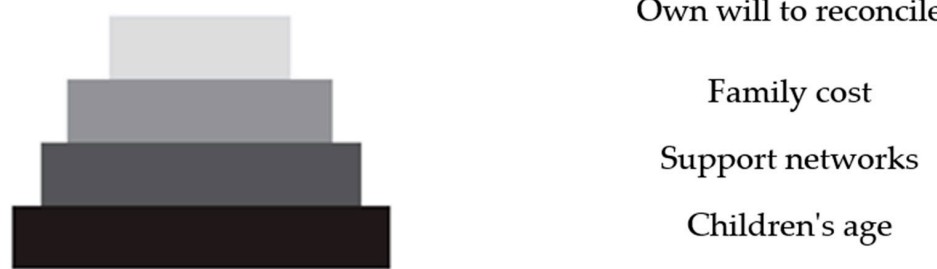

**Figure 2.** Relevant aspects involved in reconciling work and family; Source: Gaete-Quezada [43].

Different research shows the existence of barriers preventing women from accessing positions of leadership and responsibility [45]. Gender biases give rise to different barriers that researchers

have explained through the elaboration of concepts such as "glass ceiling", "diamond ceiling", "cement ceiling" and "sticky floor" that means: gender-based barriers identified for job promotion.

Glass ceiling: Invisible barriers that hinder many women with high qualifications and personal and professional capacity from accessing the highest levels of leadership and responsibility and promotion in them in the same conditions as men [46–48].

Huete, Brotons and Siguenza [18] make the difference between:

Diamond ceiling: It is this prejudice that prevents women from being valued by strictly professional criteria.

Ceiling of cement: prejudices of the own women, that prevent them from growing in any public sphere due to lack of references and the assumption of gender roles. In addition, the cement ceiling is produced by the existence of much more pressure on women in managerial positions than on men, even causing the abandonment of the position criteria.

Sticky ground: Tasks related to the private sphere that, according to the patriarchal culture, are related to women, making conciliation difficult.

Huete, Brotons and Sigüenza [18] also studies the present's evidence of wage disparity, vertical segregation and the greater job insecurity suffered by women in the hospitality subsector in Spain.

The literature offers various explanations for the low presence of women in senior management positions and on boards of directors, causes or reasons [49] If the company examines that the entry of a woman to the board of directors is counterproductive for its proper functioning, the individuals who decide on the composition of the boards of directors, would not provide equal opportunities for women simply because they are. There is discrimination due to erroneous beliefs regarding the capacity of women; they conclude that the larger the size of a board of directors, the easier it is to find a woman in it, although this presence grows less than proportionally with size, which can be considered an indication that the presence of women in many boards of directors is merely testimonial [50].

The percentage found in most of the studies of female directors (around 4%), is substantially lower than that of female managers, which seems to indicate an under-representation of women in the boards [50]. The 50% of Spanish companies did not have a woman on their board in May [51]. That is, once again the woman meets that already well-known glass ceiling.

### 1.2.1. Causes: Maternity

For companies, maternity leave or leave can develop considerable disorders. This becomes more important in those small departments of high responsibility where the presence of women is almost total, and the possibility of multiple pregnant women can coexist at the same time. However, in those positions with no responsibility and more manual jobs, the replacement is easier, since they do not require a high level of training [52].

The involvement by women of family responsibilities constitutes the clearest barrier to access and promotion to managerial positions [50].

Other causes that are behind this lower proportion of women with the experience required to be a counselor are family responsibilities that in many cases interrupt the development of professional activity and require dedication to home care [50].

Among couples with children inequalities are accentuated, the employment rate (full-time equivalent) for women is 52%, compared to 75% for men. The gender gap is five times greater among couples with children than that of childless couples [53].

The age of the children is one of the aspects to highlight and especially in the importance that the age of the children has in a woman's directives; at a younger age, the difficulties of work-family reconciliation increase more, which tends to decrease when the children become more independent. This could be supplemented with strategies in the organization of work: more flexible hours, teleworking, temporary reduction of hours, paid breaks, facilities for breastfeeding at work, etc. Also, the relevance of the age of the children determines the size of the work-family conflict [46].

However, in the Spanish hospitality sector, women earn less than men and are less represented in management positions. In this regard, family responsibilities are recognized as the main cause that keeps women away from management positions, supporting an unequal distribution of responsibilities at home [54]. Thus, the social consensus on the role it should assume in the family environment (care, motherhood, etc.).

Family duties still have consequences in women's job choices, especially in the positions on the board of directors of the hotel industry, where there is a great commitment, a great dedication and also must have availability to travel; therefore, flexibility is a basic requirement for working in tourism [55].

One of the main factors to achieve a balance between work and personal life may be one of the main factors that make a glass ceiling [19]. In the tourism sector there is a glass ceiling for women depending on where they are in their professional cycle, there is a culture of long working hours, geographical mobility and the choice that women have to make between work and personal life [51].

Self-imposed barriers have to do with family and domestic responsibilities, which have a higher priority for women, also the balance between work and life, there are also barriers at work such as lack of planning, stereotypes about women, considering that women are less flexible than men because the family priority prevails [19,56,57].

For all this, it is affirmed that women are assigned certain jobs: receptionists, administrators, housekeepers, shop assistants, housekeepers, etc.; this is known as horizontal segregation. It is also observed that women make use of part-time and temporary jobs to be able to be included in the world of work without neglecting the home and the family. We can affirm that women are the ones who mostly see the need to reconcile work life with life family/personal as a priority because gender roles and stereotypes fall upon them when it comes to being included in the tourist labor market [58].

In the area of what is called family-work-personal reconciliation, there is a home-career conflict in which the typical sexual role of men places them outside the domestic sphere, while that of women places them within the home [59]. Table 2 shows the work-family problems citing authors that have studied it.

**Table 2.** Work Family Balance.

| Authors | Work-Family Problems |
|---|---|
| Chen and Powell [60]; Greenhaus and Beutell [61] | A person has a fixed amount of time and energy to spend on life roles; Participation in a role depletes the resources available to others, as a result of which people who have multiple roles experience conflicts between roles due to demands and expectations of competing roles, such as work-family roles. The roles are incompatible. |
| Kossek, Hammer, Kelly and Moen [62] | Conflicts and stresses of work, family and personal life are growing management and public health problems affecting employees, employers and families worldwide. |
| Michel, Kotrba, Mitchelson, Clark and Baltes, [63] | Indicate that job stressors, job characteristics, variety of tasks, job autonomy, among others are antecedents of work-family conflict. |
| Pérez, Jiménez, Garcés and Sánchez [64] | Conflict situations are especially pressing, and middle managers must be able to identify physical symptoms before they generate illness, absenteeism or excessive job turnover. There must be a leading approach to family-responsible work relationships. |
| Süß and Sayah [65] | Work challenges the individual balance between itself and personal life when it comes to long work hours and absences from home. |
| Smidt, Pétursdóttir and Einarsdóttir [66] | Work-family balance is especially unfavorable for women because they are negatively affected by spending more time on family obligations compared to men. |
| Richter, Näswall, Lindfors and Sverke [67]. | Relevance of the age of the children to increase the work-family conflict for women |
| Tsionou and Konstantopoulos [68] | There is a need to study the association of labor and family conflicts with the school performance of children, as well as the satisfaction of the life of the spouse and children related to psychological stress and negative emotions. |

Source: Own elaboration based on Gaete–Quezada [46].

### 1.2.2. Causes: Extended Workday

An extended workday is another factor that affects work-family reconciliation, since it creates additional difficulties for women when it comes to assuming a managerial position, which requires family collaboration, for example when women have to carry out business trips. The greater number of hours of work that a manager must dedicate to their functions is also a variable analyzed by some studies on CTF [69].

### 1.2.3. Causes: Social

If we look at the proportion of men and women who respond to the degrees of obligation to care for people in their charge, the greater the degree of responsibility needed, the greater the proportion of women who care for other people [53].

The main brakes that are perceived are in general of a cultural and invisible type, which are carried by social transmission [53].

Among the reasons that women are less commonly hired to carry out these positions would be the existing occupational segregation that tends to place men in positions of financial or more technical content within the production process [70].

Gimeno and Rocabert [71] argue that women who achieve success do not enjoy it in the same way as men because they have to relegate family life for a time. This may be associated with the professional success of the mother or wife causing a rejection among the other members of the family, causing the woman contradictory and guilty feelings. In this way the traditional role of women as mother and wife is reinforced, linked to the family home in which all their priorities must be concentrated.

If women resist this imposition, then inter-role conflicts related to lack of time and stress occur. The domestic responsibilities culturally linked to the female role, together with a different socialization according to gender, causing women to have to deal with endless hours, parties and days in male work spaces, incompatible with the other traditional roles of mother and housewife [54].

## 2. Materials and Methods

### 2.1. Materials

Hotels are immensely important on a nationwide basis and represent a large economic sector, according to a report drawn up by Horwath Hotel, Tourism and Leisure (HTL) in 2018, in which the situation of the hotel trade was analyzed in 14 countries in continental Europe. Spain is one of the countries with the most hotel chains, the third largest supplier, after France and the UK according to Horwath Hotel, Tourism and Leisure (HTL) [72].

Moreover, according to this report, Spain ranks number one among European countries in terms of national hotel chains, ahead of Italy and Germany.

These hotel chains were chosen for the analysis of positions in boards of directors, as Spain is one of the European countries where large companies such as these are plentiful.

The degree of female participation was found both on executive committees and in general management. In a similar vein, data was gathered on whether the hotels put gender equality into practice by analyzing the number of women who perform this type of work.

The criteria were chosen from the supply data (number of establishments and positions) and was ranked from surveys in which 120 companies participated and provided data, which was used to draw up the hotel chains ranking [73–81].

In total 23 hotel chains were observed in order to obtain the sample, among which were: Meliá Hoteles Internacional, NH Hoteles, Grupo Barceló, RIU Hotels and Resorts, Eurostars Hoteles (Grupo Hotusa), H10 Hoteles, Iberostar Hotels, Hoteles Globales, Best Hoteles, Hoteles Playa Castilla, Catalonia Hotels and Resorts, Princess Hotels, Palladium Hotel Group, Grupotel, Paradores, Hipotels, Allsun Hotels, Grupo Blue Bay, Accor, Grupo Piñero, Be Live Hotels, Vincci Hoteles and Ilunion Hoteles. Among these, the first five that appeared on the list of the most widespread hotel chains in

Spain were chosen, as shows Table 3 with the number or Establishments and number of Rooms for top five hotel chains established in Spain.

Meliá Hotels Internacional: this company is considered to be the most important chain of all due to its global size. In 2019 it owned 329 establishments and supplied 83,018 beds. Its managerial structure is made up of: a CEO and Vice Chairman, Chairman of the Board and 9 Directors. There are 11 board members in total, 3 of them are female [82].

NH Hoteles: In 2019 NH has owned a total of 369 establishments with 57,356 beds [81]. It operates in 31 countries, but Spain is where most of its capital investment is located [76].

At NH the Boardroom is composed by a President, Chairman and 7 other members, with 8 of them being men as well as one woman [83].

Barceló Hotel Group: In 2019 it took third place in terms of number of establishments and rooms, with a total of 55,944 beds and 251 establishments [81]. Barceló Hotel Group is the hotel division of Grupo Barceló.

The top management of Grupo Barceló, which has been owned by the Barceló family for three generations, is represented by the current co-chairpersons and a director, all of whom are male. All of the total of 15 persons managing this company are men [84].

RIU Hotels and Resorts: In 2019 the Majorca based company RIU owned 95 hotels throughout the world, according to data published by Hosteltur and provided a total of 45,648 beds [81].

The boardroom is made up of 7 men (there were 7 men and one woman, but woman left in February 2019) [85,86].

Iberostar Hotels & Resorts: which provides 32,404 beds in total, distributed over 102 establishments, takes fifth place in the number of rooms provided, but is 4$^{\text{th}}$ in the total number of beds [81]. Its boardroom is made up of 7 men and 3 women [87].

It must be taken into account that at every hotel chain, the executive committee is made up of a different amount of people, who hold different positions and each company has its own organizational chart.

*2.2. Methods*

The methodology applied for this research, is the one proposed by [85] cited by Campos–Soria, Marchante–Mera and Ropero–García [17], published in the International Journal of Hospitality Management. They say: "European researchers generally use the methodology proposed by Hakim (1992) [1] to measure the percentage of men and women in each occupation, with the aim of classifying them into occupations dominated by women, by men or gender-integrated people. Reference [1] proposed an approach that focused on gender dominated and gender-integrated occupations. Thus, the representation coefficient for each occupation is defined for both genders. The coefficient of female representation (CFR) in occupation i is obtained by dividing the female share of employment in this occupation (Fi/Ti) by the female share of total employment (F/T), where Fi and Ti are the number of women and the total number of workers in occupation i, respectively, and F and T are the number of women and the total number of workers in the sample, that is: [(Fi/Ti)/(F/T)]. Similarly, the coefficient of male representation (CMR) in each occupation is calculated as [(Mi/Ti)/(M/T)], where Mi is the number of men in occupation i and M is the total number of men in the sample.

Hakim's [1] methodology has been used to classify occupations, and female-dominated, male-dominated and integrated occupations can be used.

The concepts are:

$$CFR = [(F_i/T_i)/(F/T)] \tag{1}$$

$$CMR = [(M_i/T_i)/(M/T)]. \tag{2}$$

**Table 3.** Ranking of the top five hotels chains established in Spain.

| | HOTEL CHAINS | 2010 * | 2011 * | 2012 * | 2013 * | 2014 * | 2015 * | 2016 * | 2017 * | 2018 * | 2019 * |
|---|---|---|---|---|---|---|---|---|---|---|---|
| 1 | Meliá Hotels International | 307/ 77.635 | 309/ 77.821 | 306/ 77.996 | 302/ 77.894 | 308/ 79.000 | 309/ 82.283 | 314/ 83.252 | 312/ 80.305 | 321/ 80.861 | 329/ 83.018 |
| 2 | NH Hotel Group | 394/ 58.911 | 400/ 59.885 | 395/ 58.885 | 386/ 58.168 | 369/ 57.785 | 382/ 59.047 | 381/ 58.714 | 379/ 58.676 | 385/ 59.682 | 369/ 57.356 |
| 3 | Barceló Hotel Group | 183/ 47.153 | 109/ 43.081 | 108/ 43.081 | 106/ 44.435 | 104/ 45.277 | 104/ 44.490 | 109/ 32.770 | 229/ 50.486 | 247/ 54.219 | 251/ 55.944 |
| 4 | Riu Hotel &Resorts | 106/ 40.083 | 163/ 37.778 | 140/ 37.778 | 140/ 37.578 | 140/ 37.380 | 107/ 33.379 | 93/ 42.291 | 94/ 43.873 | 90/ 42.497 | 95/ 45.648 |
| 5 | Iberostar Hotels &Resorts | 100/ 33.000 | 92/ 36.000 | 89/ 30.063 | 88/ 30.181 | 76/ 26.806 | 77/ 27.262 | 78/ 27.551 | 83/ 28.921 | 100/ 31.824 | 102/ 32.404 |
| | TOTAL | 1.090/ 256.782 | 1.073/ 254.565 | 1.038/ 247.803 | 1.022/ 248.256 | 997/ 246.248 | 979/ 246.461 | 975/ 244.578 | 1.097/ 262.261 | 1.143/ 269.083 | 1.146/ 274.370 |

* Number of Establishments/Number of Rooms; Source: Prepared from the authors of [73–81].

## 3. Project Data

Data collection refers to quantitative analysis. In this case, the different positions on the board of directors at the hotels have been numbered and the total number of positions and the number of men and women holding each position on the board has been counted.

*Focus of Research*

The aim of this research is to find the proportion of women occupying these senior positions and from that point compare them to those held by men in order to confirm that gender inequality exists:

(a)   The proportional ranges of women in executive committees or managerial boards (minimum and maximum).

(b)   The proportional ranges of men in executive committees and general managerial teams (minimum and maximum) from calculating (a) and (b) as these are directly obtained by subtracting the proportion of women from the total which = 1.

## 4. Results

The data on the sample is shown in Table 4, which has been drawn up according to the variable under consideration:

**Table 4.** Gender distribution of members of the executive committees of hotels.

| Company | Number of Establishment (2019)/Number of Rooms | TOTAL Number Board's Members | Men | CMR | % Men | Women | CFR | % Women |
|---|---|---|---|---|---|---|---|---|
| Meliá Hotels International | 321/83.018 | 11 | 8 | 0.84 | 15.38 | 3 | 2.03 | 5.77 |
| NH Hotel Group | 369/57.356 | 9 | 8 | 1.03 | 15.38 | 1 | 0.83 | 1.92 |
| Barceló Hotel Group | 251/55.944 | 15 | 15 | 1.16 | 28.85 | 0 | 0 | 0 |
| Riu Hotels & Resort | 95/45.648 | 7 | 7 | 1.16 | 13.46 | 0 | 0 | 0 |
| Iberostar Hotels & Resorts | 102/32.404 | 10 | 7 | 0.81 | 13.46 | 3 | 2.23 | 5.77 |
| TOTAL | | 52 | 45 | 4.99 | 86.54 | 7 | 0.13 | 13.46 |

Source: Prepared by the authors from [81–87].

When the coefficient of female representation is greater than the unit, then females are over-represented in the given occupation. If the coefficient is lower than the unit, then they are under-represented. Following this methodology, occupations are grouped into gender-integrated, female-dominated and male-dominated occupations. Hakim (1992) [1] considers that a job is integrated when the participation coefficient of women in such an occupation (Fi/Ti) lies within a range ±10% of the ratio of women's share of total employment (F/T). A job is female-dominated when the coefficient is higher than this range, whereas a job is male-dominated when this coefficient is lower than this range". So, we can conclude that all of the women ratios are then lower than 10%, meaning the jobs are male-dominated. There are also two hotel chains that has a 0% ratio, showing that there are no women on the board.

It is also worth commenting on the first report made by the Turijobs portal in February 2018 [88], since it has been carried out using surveys of a total of 898 professionals in the tourism sector. It concludes that 90% of women and 74% of men consider that there is labor inequality between genders, and 64% of women affirm that they have suffered labor discrimination based on gender, with the most affected job categories being middle managers (74%) and executives (70%).

The trend is that women are underrepresented on the Boards of Directors of tourism companies, while they earn 15% less than men in the tourism sector. The Meliá Hotels International hotel chain, in its 2009 Report, states the following: "The tourist activity has an intrinsic problem when it comes to reconciling work and family life that arises from the same business operations: complex work schedules and seasonality are their basis" [89].

The data comes from the top five hotel chains, which have a total number of 1146 Hotels and 274,370 beds. If we study the number of women that in at the boardrooms of the total of the top five hotel chains, 7 are women, while the male total is 45, meaning the percentage of women is 13.46%. That is an insufficient amount to reach the CNMV's recommendation for 2020, which set a target of 30% for female representation on the boards of directors of companies by 2020. The current figure is at 20.3% in the Continuous Market, according to data from the report Women on the Boards of Listed Companies [90].

It is important to increase the quotas of women on the boards of directors and to explicitly address the educational issue of gender equality [41].

In the statistics of the last few years, it can be seen that the number of women in middle management has increased in the last decade (a 1.4% plus than 2009); however, the same has not happened with the occupation of management positions [15].

These inequalities are reflected in analyses, reports and research carried out, where women in general are not well represented in Spanish companies. The low representation of women on boards could be considered as a very important and caused by the fact that in the Spanish labor market, there are a series of difficulties, obstacles or obstacles that hinder or impede the professional development of women, which is also subject to discrimination in terms of remuneration and also in access to highly qualified positions, known as the Crystal Roof [91].

Spanish companies also show difficulties in recognizing the role of women, emphasizing issues related to gender stereotypes. This is due to a structural problem in the labor market fueled by a culture that points to beliefs, prejudices and some values that are also present in the company and in the family [92].

Since the 1970s, the incorporation of women into the labor market has been observed gradually, but the same increase in the participation of women in management positions and, especially, in the highest decision-making bodies of companies has not been observed [91].

Many empirical studies say there is a positive relationship between female managers and financial performance [90], while the incorporation of women on the boards of directors has a direct influence on the productivity and creativity of the governing bodies [45]. Women add value to boards of directors, as they have unique perspectives on experiences and work styles. Women's communication style is more participatory and process oriented, and they are oriented towards the needs and interests of consumers. There is a relationship between the presence of women on a board of directors and the number of women in executive positions; therefore, this serves as an example for other women. The inclusion of women on the boards of directors is also well received by clients and investors, with the absence of gender diversity seen as causing negative publicity for organizations. Another positive aspect is associated with the performance of a company's shares, where the companies run by women have the greatest benefits [93].

It is also worth commenting on the first report made by the Turijobs portal in February 2018 [88], since it has been carried out from surveys of a total of 898 professionals in the tourism sector. It concludes that 90% of women and 74% of men consider that there is labor inequality between genders, and 64% of women affirm that they have suffered labor discrimination based on gender, with the most affected job categories being middle managers (74%) and executive profiles (70%).

The trend is that women are underrepresented on the Boards of Directors of tourism companies, and they earn 15% less than men in the tourism sector. The Meliá Hotels International hotel chain, in its 2009 Report, states the following: "The tourist activity has an intrinsic problem when it comes to reconciling work and family life that arises from the same business operations: complex work schedules and seasonality are their basis" [94].

## 5. Conclusions

Tourism is evidently a key industry in the Spanish economy. At the same time, hotels are the largest category for accommodation. Women frequently choose careers in tourism before other ones,

at least in comparison to men. However, the status quo is one in which women face more barriers to accessing and obtaining positions in executive committees and general management, and find it harder to go up the corporate ladder in tourism than they do in other industries.

After having implemented the methodology based on Hakim [1], we conclude that women ratios are all lower than 10%, (being 5.77%, 1.92%, 0, 0, and 5.77%) meaning jobs are male-dominated in all of the top five hotel chains in Spain. There are also two hotel chains that have a 0% ratio, showing that there is no women on their boards.

So, finally, the results of all the hotel chains presented in the study lead us to the same conclusion, which supports the data collected. The objective of this work was to study in depth what happens in the area of gender in the direction of the boards of directors of large hotel chains in Spain, for which we carried out an analysis of these 5 hotel chains throughout 10 years, from 2010 to 2019, using data that have been recently published. On the one hand, we found that the distribution of men and women as leaders in the various boards of directors does not represent the minimum required by the LOIMH Law in Spain [24] due to the famous "glass ceiling", which forces women work within the tourism sector in the jobs with the lowest level of responsibility. As such, you can find a high female participation rate in the areas of cleaning rooms, administrative positions, telephone operator and assistant. There is gender equality in reception and reservation positions, but this is not the case in positions of high responsibility, with a total of 51.66% of these positions being male.

There is a difference between the distribution of men and women in different work areas in the tourism industry, where the concentration of each gender varies according to the different levels of responsibility associated with it. This difference is important to know, since it means that there will be different causes and consequences for these trends.

## 5.1. Position of Women as Part of the Executive Structure

Women are becoming ever more involved in decision-making, but mainly in middle management positions. Their participation in boardrooms is still limited, which is observable in the size of the sample chosen.

## 5.2. Gender and the Employment Situation in the Tourism and Hotel Industry

Even though, according to the population censuses, there have always been more women than men, most are unemployed and only a minority are in the workforce and are employed. Males predominate in tourism, but only very slightly; this industry is much more balanced in terms of male and female participation, and is considered to be a mixed gender one. In the accommodation subsector, however, women have always predominated (making it a feminized sector), but only at lower echelons (the higher positions are occupied by males). There is horizontal and vertical segregation, with more women than men concentrated at low levels and their numbers in management have hardly increased. Exactly the same is true for hotels. These findings correlate with a fall in female employment in tourism, and in the hotel trade. This gap is due to the time women spend at home. Women tend to be concentrated in more precarious work and do most unpaid tasks; in short, it could be claimed they work more than men over the course of their lifetimes.

In a similar vein, the main barrier they face is the work-life balance, as many authors and the executives themselves (both men and women) have explained. In the tourism industry it is far more complicated to achieve this work-life balance than elsewhere, since geographical mobility, full-time availability and flexibility are all required.

The lack of women in top managerial posts is something which affects the whole of society and the way companies behave towards women, which is why part-time contracts are so widespread, since women are associated with being full-time homemakers and often housework is not shared with their male partners.

The sectors in which female directors predominate are associated with care, feminized sectors, such as education and health centers.

Lastly, it must be stressed that the aim of this research is not to portray women as victims, but to show data as fairly as possible, whilst being realistic, and to analyze data published by companies. There is no intention to promote a type of feminism in which women are given special treatment; the article simply aims to promote gender equality. Statistics do not´t lie and most of them are indicative of a situation in which there is still room for improvement.

*5.3. Women, Hotels, Governments and Society Recommendations*

The four most important points which change, and consecutive improvement hinge upon are: women themselves, companies, governmental institutions and society as a whole. These variables must be interconnected and coordinated so that they all move in the same direction and change comes swiftly:

- Women: Women need to shoulder the main responsibility; they must be willing to take on high responsibility managerial positions, despite other considerations. Women need to demand more from themselves and companies and show the same determination and sacrifice that men do to reach managerial positions.

They must believe they are highly capable of taking on responsibility and new challenges and strive to seize opportunities just as males do, but not all women are willing to do this. Men tend to take it in their stride when requesting this type of promotion, while often women believe they will not be able to cope with the workload (work and family) which comes with such positions.

They need to stand up for their right to be treated equally to receive promotions. They must make a stand and join forces to demand they be treated equally in terms of employment opportunities, salaries and high-responsibility positions and ensure that the so-called "promises" in corporate policies on diversity are something more than just lip service.

They need to move away from their stigmatized, stereotypical roles in society, which are very often spread by them themselves, even though balancing a managerial position with family life is becoming more and more viable.

*5.4. Change in Women Attitudes towards Management*

A series of concrete actions must be carried out to improve the situation of women in the tourism sector, in three different areas: private, business, and institutional and public administration [48]. The self-imposed barriers are made when women create barriers to their own professional growth. Women are expected to behave like women, but traditionally female behavior is not considered to be desirable in a leader. For example, ambition is not considered a feminine trait. Our results may also be executives who have been able to reach managerial positions have had to challenge traditional gender roles both at work and in their personal lives. Our study also shows that the hotel industry is a highly traditional sector, in which masculine values prevail. In Spain, most of the hotel companies have been operating for fifty or sixty years and the organizational culture is a reflection of the culture of that period. This contributes directly to the pattern of employment in the industry, in which women are appointed to traditionally female roles [19]. The self-imposed barriers of women are also known as a "cement roof" triggered by the prejudices of women themselves, which prevent them from growing in any public sphere due to a lack of references and the assumption of gender roles. In addition, the cement ceiling is produced by the existence of much more pressure on women in managerial positions than on men, even causing the abandonment of a position [95].

- Hotels: it is crucial for hotels to recognize that major differences exist. They must start by raising awareness and then by putting measures into practice, not just paying lip service to this problem.

To a large extent, hotels largely ignore Equality Plans in the LOIMH [24]. This is actually detrimental to companies, as good leadership comes from great talent and managerial skills, which is something

inherent in both genders. Also, "the more open-minded" the company is in this respect, the more value, better ideas, results and future perspectives it will have.

In corporate policies, it is evident there is great deal of awareness on this issue, but there are few real data. Hotels do not support research into the current employment situation in this sector, but they ought to do so.

Another solution which may bring about improvements is to hire mentors to encourage women to aim more for managerial positions. This would provide both a reference figure and adequate professional development.

In the information reviewed, men spend more time working for companies than women, which is something that should be changed in organizational charts (dinners, business trips, meetings, etc.). Perhaps such positions are attractive to men, but not to women due to the complications often entailed in balancing work and family life.

Companies need to show they are committed to equal opportunities and that they have rigorous controls for assessing results from their gender equality policies. In fact, this is already part of their competitive strategy in so far as more and more companies are supportive of and committed to diversity, meaning those which are not will be left behind and viewed negatively.

Finally, hotels must be called on to hire women through recruitment processes, although these still contain barriers to promotion, which results in practically all those hired being men.

- Government: it has a lot of influence on to what extent females hold executive positions, as seen in the improvements in the public sector.

Perhaps, this is not the best solution, but for women to reach managerial positions, some specific measures are required. Governments need to enforce equal opportunity improvements on private companies using legislation and to fine companies which do not keep up to date with improvements in this respect. In this way, if they do not invest in greater development and diversity, such as by duly implementing an Equality Plan, then they must be sanctioned so that they will think twice before making this mistake again, as the reputation of companies that do not push for equality will damaged. Perhaps, legislation may be the most effective way of carrying this out.

The law can and must improve in terms of regulating more effective quotas for directors and those on executive committees, and cannot permit horizontal segregation, since many women have patently proven themselves to be just as ambitious as men.

Maternity and paternity leave should be put on par in order to have real equality in terms of the work-life balance since birth (bearing in mind that biologically the pregnancy is more important for the woman), since if a woman today can balance her managerial position with her family life, it is often due to her partner and not thanks to any legislation or corporate initiative.

- Society: In general, society tends to behave as most people do, while deep-rooted and outdated customs and ways of thinking still persist, with roles and social stereotypes in which women are associated with being the weaker sex and caring for others.

The solution is to change how society views women, so that they do not feel there are barriers to them obtaining higher positions and this must be perceived as being "something normal". This way of thinking must, in turn, bring about a change in attitudes at companies and in men and/or the spouses of female directors or those who are aiming for this professional goal.

Hotel employees must go even further and call for greater quality of life when performing these demanding positions, and therefore family and childcare must be distributed on a more equitable basis, since how society as a whole behaves, has a considerable influence on companies.

The fundamental factors influencing this phenomenon have been considered, but there are other secondary aspects which also affect it such as training centers for tourism, education, levels of income, searches for opportunities or the media, to mention just a few.

**Author Contributions:** M.J.C.-S. Principal Investigator; C.C.R. Second Investigator and E.R.R. Investigator and Data Collection. All authors have read and agreed to the published version of the manuscript.

**Funding:** This research received no external funding.

**Conflicts of Interest:** The authors declare no conflict of interest.

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
