# Peer review of "Why Is the Spanish Hotel Trade Lagging So Far Behind in Gender Equality? A Sustainability Question"

_sustainability, doi:10.3390/su12114423_

Round 1

Reviewer 1 Report

The topic of the paper is interesting and relevant. However, the relationship between gender equality and social/economic/ecological sustainability could be more clearly defined.

The legal framework is presented in the discussion section, it would be more relevant in the introduction as background information for the study.

The information in the material and methods section is very detailed, it could be presented in a table rather than text to make it more clear for the reader.

There are several questions posed in the introduction, but not all of them are addressed in the paper. For example, a question related to the economic success of hotels run by women is mentioned, but there is no information of this in the data covering the five hotel chains. As only the proportion of women in the executive committees of the hotels is reported, no conclusions can be made of the possible variance in sustainability.

The problems of work-family balance is seen as one of the main barriers for women to enter management positions. In order to highlight this perspective it would be relevant to present information about the family status of women who have made it to the hotel executive positions: are they married or single, do they have children?

In the recommendations section, change is proposed in individual women's attitudes towards management tasks. There is however no data or previous research presented in the paper about what the variation of attitudes among women is.

Author Response

Response to reviewer. Please see the attachment.

You can see attached a word with the responses, thank you so much for your interesting notes about the paper.

Best regards

Reviewer 2 Report

This is an interesting article on a topic of high interest, relevant for practitioners, policy-makers and scholars, and it is my pleasure to review it.

Formal- The paper must be reviewed in formal terms, page layout, references in the text. Final references have to be drafted in mdpi style. Figure 1 is truncated, insufficiently explained and practically, useless.

Content- Although the idea is interesting and the interpretations of the situation in legal, organizational terms are suggestive, the quantitative analysis is simplistic, i.e the whole research is underdeveloped. Chapter 4. Results, is obviously schematic, but, paradoxically, at the same time, abundant in redundant figures, which have already been presented in the previous table and the figure. In fact, Table 2 and Figure 1 show the same data.

The data refer to one or two years (2018, 2019). Even if these figures are eloquent, the image is incomplete, it need time series, evolutions, structures and trends to identify a phenomenon, even if, at first glance, it appears obvious and rooted in history.

The fact that women are under-represented in the management structures of large hotel chains is obvious, but we consider the author had to address more deeply the causes: organizational, experience, expertise, seniority, discrimination, traditionalism, power conservation, etc. by using statistical data, interviews, questionnaires, data that can be interpreted, adding value to these deductions.

Author Response

Dear reviewer, please see the attachment.

Thank you so much for your work, your comments are very interesting for us.

Best regards

Round 2

Reviewer 1 Report

The paper has improved: the relationship of gender equality and sustainability has been defined and the work-family balance has been highlighted as the main barrier for women to enter management positions.

The information in the data and methods section is still very detailed, but the table in the result section helps in perceiving the main points.

As text has been moved between different parts, the numbering of references is no longer in numerical order and needs to be checked.

Author Response

Dear reviewer,

We have completed the references and checked them all, we have made a new table for see the results properly.

Thank you so much for your help.

Best regards, 

Reviewer 2 Report

The paper was enriched with additional explanations, often necessary and suggestive. However, the paper still suffers from major problems, related to the depth of research and structure:

  1. The phenomenon is not studied in time, there is no temporal analysis of it, being limited at only 2 years (2018-2019). Although the problem of under-representation of women in business management is well known (social, global), this had to be documented for a long enough time to certify the chronic nature of this problem, also in tourism industry. Then it had to be analyzed in relation to other consequences - efficiency, profitability, turnover. A relative equality of male and female managers is based, at least in for-profit companies, on relatively similar performances. If not, it is very easy to enter in the area of ​​propaganda and unsubstantiated statements.
  2. In terms of structure, curiously, the authors preferred to gather a lot of additional explanations, a kind of ad-hoc literature review, the Section 5. Discussions and conclusions (see, for example, Table 4). This section is usually intended analysing the results of the own research, interpreting and reporting them to the main findings in the field.

Author Response

Please, see the attachment,

Thank you so much for your recommendations. 

Best regards, 

Round 3

Reviewer 2 Report

In this last form, the paper is better argued, and the formal aspects were improved.